# Influence of variation in the volumetric moisture content of the substrate on irrigation efficiency in early potato varieties

Anna Jama-Rodzeńska[1☯]*, Amadeusz Walczak[2☯], Katarzyna Adamczewska-Sowińska[3☯], Grzegorz Janik[2☯], Izabela Kłosowicz[4☯], Lilianna Głąb[1☯], Józef Sowiński[1☯], Xinhao Chen[5☯], Grzegorz Pęczkowski[2☯]

**1** Division of Plant Production, Institute of Agroecology and Plant Production, Wroclaw University of Environmental and Life Sciences, Wroclaw, Poland, **2** Instiute of Environmental Protection and Development, Wroclaw University of Environmental and Life Sciences, Wroclaw, Poland, **3** Department of Horticulture, Wroclaw University of Environmental and Life Sciences, Wroclaw, Poland, **4** Students Scientific Association of Melioration, Hunan Agricultural University, Changscha, China, **5** Hunan Agricultural University, Changscha, China

☯ These authors contributed equally to this work.
* anna.jama@upwr.edu.pl

**Data Availability Statement:** All relevant data are within the manuscript and attached figures.

## Abstract

Potato is a plant with high water requirements. This factor affects not only the weight of potato tubers but also their quality parameters. In order to achieve quantity and quality goal, it is helpful if we apply the principles of precision agriculture, which also contributes to sustainable management of environmental resources. Accurate identification of the water requirements of crops is the basis for determining optimal irrigation doses and dates. After their application, it is possible to assess the effectiveness of irrigation treatments and their impact on the air-water conditions in soil with a root system. The aim of the presented study was to analyse the influence of volumetric soil moisture diversity on the vegetation of early potato varieties. Two potato varieties were subject to investigation: Denar and Julinka. Pot experiments were carried out at the Department of Horticulture of Wroclaw University of Environmental and Life Sciences. Three variants were analysed: one with a low water content in the soil (pF 2.7), one with the optimal water content (pF 2.5) and one with a high water content (pF 2.2). The basis for the selection of the frequency and application rate of water doses was soil moisture measured with an SM150-Kit set. Volumetric moisture was measured with a TDR apparatus. It was found that the water requirements of both potato varieties differ and increase along with the development of the aboveground and underground parts. Moreover, it was shown that the irrigation requirements of cv. Julinka are higher than those of Denar (31.4–33.0% higher), depending on the adopted variant. The research also showed that the most effective method of potato cultivation is to maintain soil moisture at a lower level. This should be taken into account in regions where the cultivation of this species uses supplementation of the water requirements by irrigation.

**Funding:** Financed by the National Centre for Research and Development as part of the project "A mobile system for precision injection irrigation and fertilization meeting the individual requirements of plants". MSINiN - Polish acronym for the project title. Grant number: BIOSTRATEG3/343547/8/NCBR/2017

**Competing interests:** The authors have declared that no competing interests exist.

# 1. Introduction

Potato (*Solanum tuberosum* L.) is the third largest crop in the world, after rice and wheat, in terms of production volume, with tuber production exceeding 330 million tonnes [1,2]. Over the last 20 years, despite a decrease in the area under potato cultivation in Europe, the crop share allocated for consumption purposes has only slightly decreased [1,3,4]. Extensive use of potato in human nutrition and starch production distinguishes it from other crop plants. The possibility for cultivation of this plant species depends to a large extent on habitat conditions, and commercial yield is determined by atmospheric, hydrological and soil conditions [5–12]. During its growing season, potato is subject to many stress factors, with drought being the main abiotic agent to which it is exposed during its growth period [13]. In the initial period of development, the plant does not manifest high water needs. However, the requirements grow significantly in the subsequent vegetation period. This should be attributed to the strong relationship between progeny tubers and mother tubers where the former, in the initial period, use winter and spring water reserves as well as water directly from the mother tubers [9,12,14]. According to Miyashita et al. (2005) [15], global warming is increasing the risk of potatoes being exposed to drought, as it causes fluctuations in the distribution and frequency of precipitations. The water requirements of this plant depend on many factors, such as potato variety, earliness group, planting time, plant structure, soil compactness or proper agricultural technology. Under optimal irrigation conditions, potato is able to produce a large number of high-quality tubers [16]. A shortage of precipitation, and consequently a decrease in soil water capacity (below 60%) in the critical period, results in a decrease in size and a deterioration of potato yield quality. Tuber yield losses resulting from precipitation deficiency range from 10 to 50%, and in some conditions even up to 70% [5,17–22]. Optimal soil moisture, alternating with water shortages, also causes deterioration of tuber quality manifested by tuber deformation.

The primary aim of this study was to compare the reaction of biometric features of potato varieties to different levels of soil moisture resulting from different volumes of water administered by an irrigation system. Additionally, the usefulness of various techniques of soil moisture volume measurement to regulate air-water relations aimed at ensuring the optimal growth conditions for plants was analysed. Moreover, the study investigated the suitability of the proposed irrigation method for maintaining the optimal soil moisture level. The existence of any correlation between the moisture content on the soil surface and that in the tuber formation area was also analysed.

# 2. Materials and methods

## 2.1. Experiment set-up

The pot experiment was carried out in 2018 at the Research and Didactic Station for Vegetable and Ornamental Plants in Psary belonging to the Department of Horticulture at Wroclaw University of Environmental and Life Sciences. Two varieties of potato were used: Denar–a very early, culinary salad, type A; and Julinka–an early general use cultivar, with rather fine-textured flesh, type B. The tubers, 90 examples of each variety, were planted under controlled conditions in a foil tunnel, into impermeable pots with a capacity of $11 \times 10^{-3}$ m$^3$ (Fig 1), one tuber per one pot. Planting was carried out at a depth of about 10–12 cm, with the eyes directed upwards. No mineral or organic fertilisation was applied. Weeds that appeared were removed by hand. Potatoes were planted in the first decade of May and harvested in mid July. The total vegetation period was 72 days. The set-up of the experiment is shown in Fig 1.

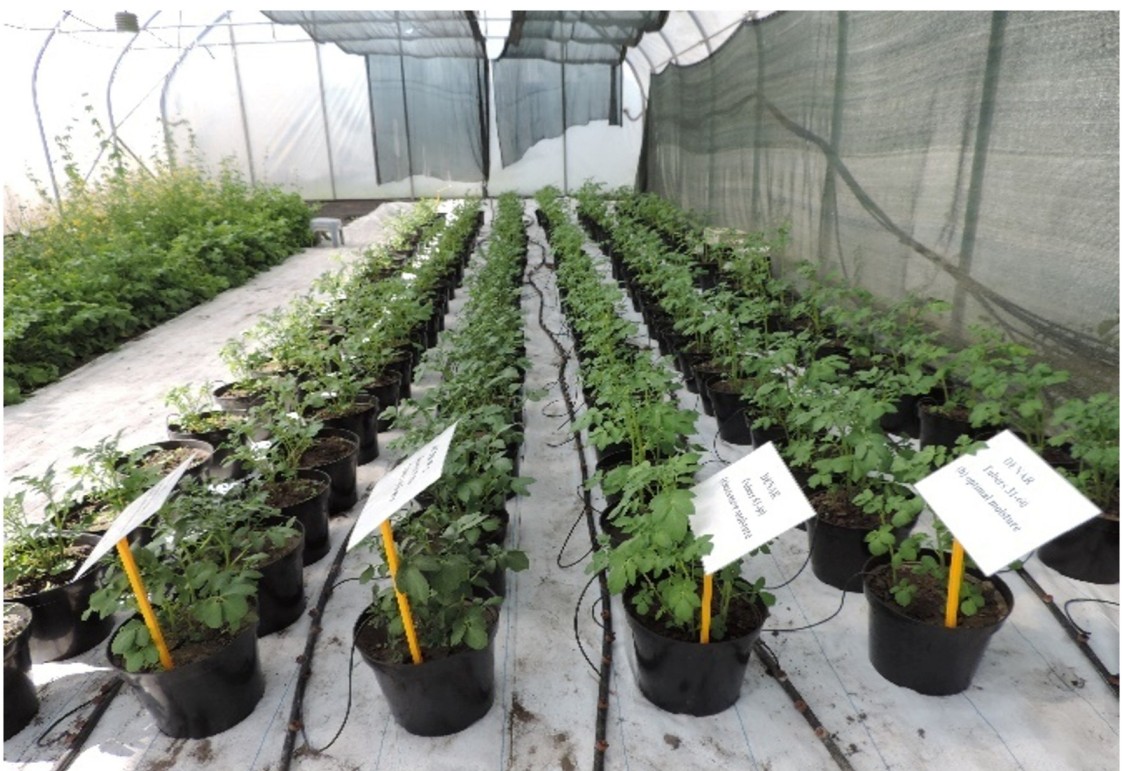

**Fig 1. Cultivation of early potato varieties in pots, equipped with a drip irrigation system.**

The pots were filled with soil whose particle size distribution was determined using a Mastersizer 2000 laser diffractometer [23,24]. This method is based on determining the particle size by using the optical diffraction phenomenon [25]. It is assumed that a monochromatic parallel beam of light is dispersed in a given medium at an angle that is greater the smaller the diameter of the particles present in the medium [25]. The results of the analysis are shown in Fig 2. The particle size distribution of the mineral parts corresponded to sandy clay. For this type of soil, the filtration factor in the saturated zone Ks is 2.88 cm/d, the humidity in the full saturated zone $\theta s = 0.321$ [–] and $\theta r = 0.109$ [–] [26]. The chemical composition of the soil used in the pot experiment is presented in Table 1.

For potato cultivation, the system used irrigation in the form of a surface drip line equipped with droplet emitters (drip pins), which periodically administered water at a rate of 2 10–3 $m^3$ $h^{-1}$. The experiment was based on the independent series method with two variable factors. The first factor was potato variety: Denar and Julinka. The second factor was the maintenance and comparison of the impact of different levels of volumetric moisture for soil in the pots. Three variants were used: low, optimal and high water content. Differentiation was achieved by the use of different irrigation doses.

## 2.2. Volumetric soil moisture content as irrigation response test

To determine volumetric soil moisture values, gravimetric methods based on the measurement of the electrical permittivity of a three-phase porous medium were used. During the experiment presented in this paper, FP/mts type TDR sensors and an SM150-Kit moisture meter were used to measure soil moisture content (Fig 3a and 3b).

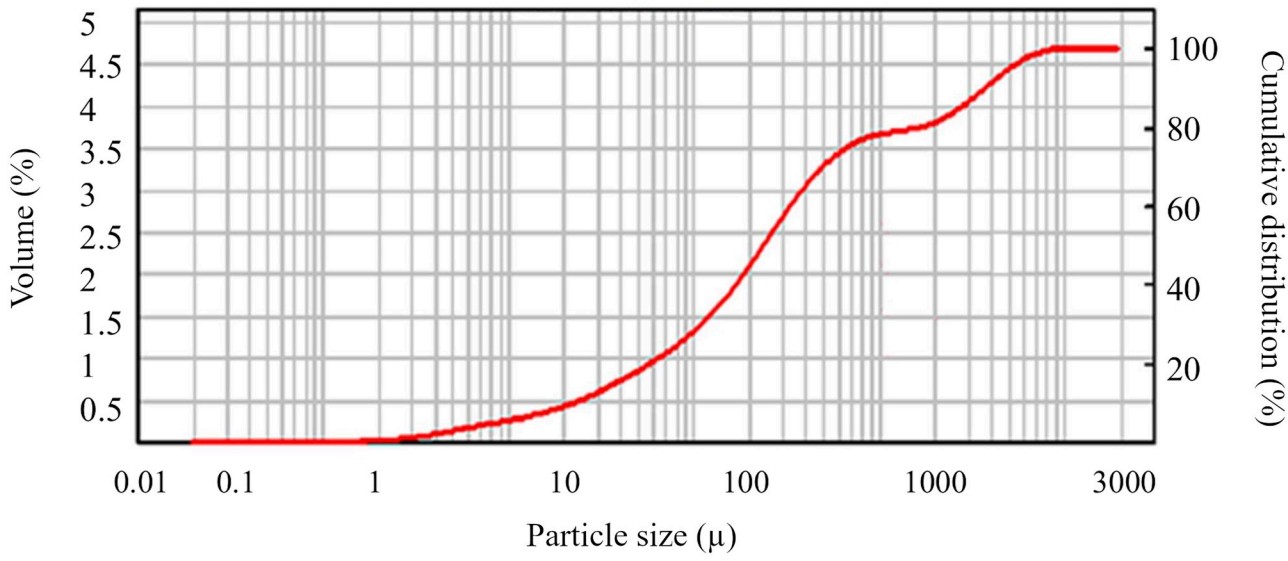

**Fig 2. Particle size distribution of mineral parts of the soil used in the experiment.**

The TDR system includes a reflectometer, a data recorder, a coaxial multiplexer, cables and probe. Sensors of this type are calibrated using water and benzene. Due to its large body diameter (d = 4.5 cm), the SM150-Kit type measuring probe was used to measure on the surface, in a line situated 5 cm from the axis of symmetry of the plant. The length of the sensor rods is 15 cm. Therefore, it is assumed that the readings represent moisture at a depth of 7.5 cm. The SM150-Kit portable moisture meter is characterised by an accuracy of surface moisture (± 3%). The portable set consists of a soil moisture sensor (SM150T) and a reading gauge.

FP/mts sensors were characterized by smaller body diameter, incorporating 2 cm longer bars (10 cm). In the case of this sensor for dry soil, the sensitivity zone is an elliptical cylinder with a height of 10.2 cm and radii of 0.3 cm and 0.4 cm. When the soil is saturated, however, the cylinder is 11.2 cm high and the base of the wheel has a radius of 1.7.

Soil volumetric moisture was determined twice a week. In this study, the TDR sensor was used to perform tests of soil moisture response to irrigation and to assess its effectiveness, while the SM150-Kit moisture meter was used to adjust the water doses [26].

In addition, a comparative analysis of both methods used in the experiment to measure volumetric moisture was performed by determining the correlation between average daily results obtained from TDR and SM150-Kit sensors. When analysing the results of statistical analyses,

**Table 1. Soil chemical composition used for pot experiment.**

| Soil characteristic | Value (unit) |
| --- | --- |
| pH | 7.16 |
| salinity | 155 μS/cm |
| Mg | 20 mg/dm$^3$ |
| P | 51 mg/dm$^3$ |
| Ca | 420 mg/dm$^3$ |
| K | 52 mg/dm$^3$ |
| nitrates | 1.31 mg/dm$^3$ |

(a)

(b)

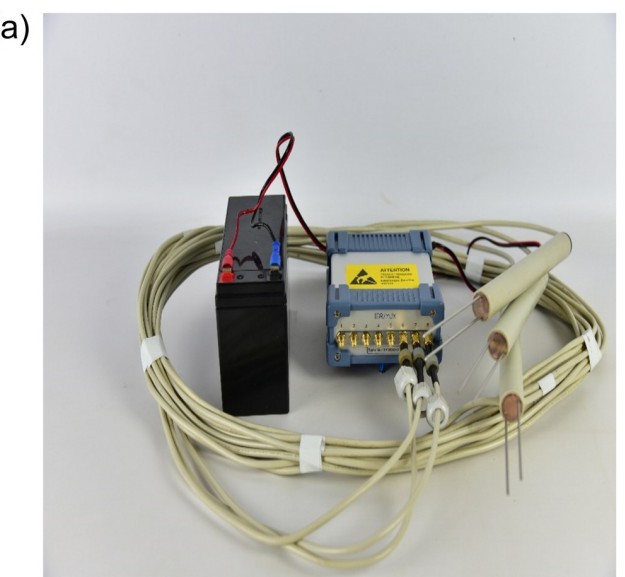
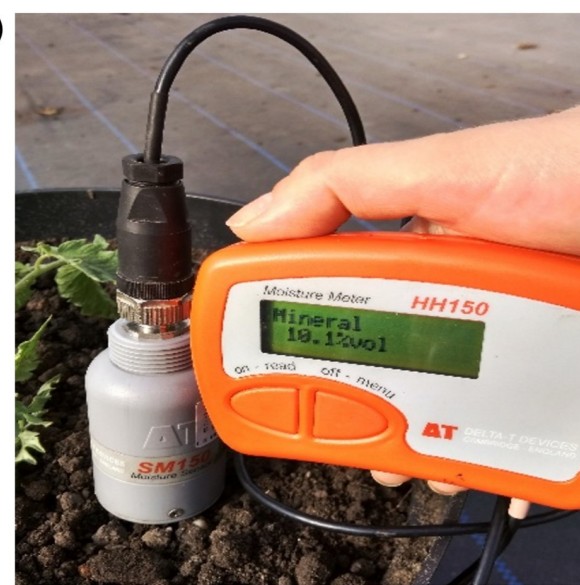

**Fig 3. Apparatus used to measure volumetric moisture in pots.** TDR apparatus with FP/mts sensor (left), SM150-Kit probe (right).

one should not ignore the different geometry of the applied rods generating the electric impulse. The analysis was performed using Statistica software (version 13.1, Statsoft).

## 2.3. Biometric determinations

Plant material was collected 72 days after planting. After washing the tubers, stolons and roots, draining and drying, they were subjected to biometric analysis. The measurements included the number of stems, length of the longest stems, number of tubers, tuber weight, number and weight of stolons, and root weight. An electronic laboratory balance with an accuracy of 0.1 g was used to determine the mass of individual plant parts. The collected results were analysed statistically using the Tukey test (post hoc).

## 2.4. Irrigation efficiency

The evaluation of irrigation efficiency analysed in the study, depending on the variety and soil moisture level in potato cultivation, was carried out concerning yield and biometric features and improvement of air-water conditions in soil with a root system. In terms of yield, the values of the WUE index were analysed [26]. This index determines the yield in relation to the volume of water given to the plant during the whole vegetation period. Due to of the experiment, it was modified to the following formula: the specificity

$$WUE = \frac{Y_i}{V_n}, \tag{1}$$

where:
$WUE$ – coefficient of efficiency of the use of irrigation water (g l$^{-1}$).
$Y_i$ – average weight of tubers from 30 pots calculated for each potato variety (g),
$V_n$ – volume of water used for irrigation throughout the growing period (l).

The second analysed irrigation efficiency coefficient indicates the size of the increase in soil volume moisture in relation to the volume of a single dose of water. The results of measurements obtained from FP/mts type TDR sensors were used to calculate the coefficient. The sensitivity zone of the sensor is an elliptical cylinder with radii of 0.5 cm and 0.8 cm and height of 5.5 cm. Such a sensitivity zone includes soil with a root system. The efficiency coefficient defined in this way was determined by the formula:

$$E_f = \frac{\theta_k - \theta_p}{V_i},$$

(2)

where:

$E_f$ – irrigation efficiency (cm$^3$ cm$^3$ ml$^{-1}$, % ml$^{-1}$),

$V_i$ – volume of a single irrigation dose (ml),

$\theta_p$ – volumetric moisture before a single dose of water is administered (cm$^3$ cm$^{-3}$, %),

$\theta_k$ – volumetric moisture after administering a single dose of water (cm$^3$ cm$^{-3}$, %).

The WUE (g l$^{-1}$) and $E_f$ (given in % ml$^{-1}$) were calculated for each cultivation and for each irrigation variant.

## 3. Results and discussion

### 3.1. Testing the adequacy of the soil moisture content response to irrigation

The distribution of volumetric moisture is the basis for the analysis of water movement in porous media–including in the substrates of cultivated plants. TDR and FDR sensors and capacitive probes [26,27,28] are used in methods based on an understanding of the electrical properties of a porous medium. The advantage of TDR is that the measurement is independent of external factors such as soil temperature and salinity. The measurement is performed automatically with a short, as little as 1 minute, time step [29–33]. The use of these types of sensors allows monitoring of the water content in soil in short time intervals, without any destructive impact or time-consuming sampling [28]. The set enables quick calibration of the measurement in a porous medium. An SM150-Kit can be used in substrates made of perlite, coconut fibre, peat and mineral wool, as well as in mineral substrates [34–37].

During the experiment, the correctness of the response of the moisture of soil with a root system to irrigation with a drip line was checked. For this purpose, the results obtained from measurements with Fp/mts type TDR sensors were analysed. As mentioned above, readings carried out in this way enable measurements to be performed with a short time lapse and, at the same time, with automatic recording. The SM150-Kit does not have this functionality. The results from pots in which both potato varieties–Julinka and Denar–were grown were analysed. Fig 4 shows a three-day period (10–12 June) of volumetric moisture readings with time step $\Delta t = 10$ $min$ for the Julinka variety. For each of the three days under consideration, the irrigation system was activated, which for the three variants (pF 2.7, 2.5 and 2.2) introduced 0.3, 0.4 and 0.5 10$^{-3}$ m$^3$ of water into the pots, respectively. For each variant, three phases were distinguished. Phase I was the moment of water being supplied by the irrigation system. Phase II was a short time period (about 0.5 h) after irrigation, and Phase III was the period until the next dose was administered. During Phase I, no rapid increase in moisture was observed for the pF 2.7 variant (dose 0.3 10$^{-3}$ m$^3$ H$_2$O). In the variants corresponding to pF = 2.5 (dose 0.4 10$^{-3}$ m$^3$ H$_2$O) and pF 2.2 (dose 0.5 10$^{-3}$ m$^3$ H$_2$O), the increments were significant and amounted to 0.02 m$^3$ m$^{-3}$ and 0.025 m$^3$ m$^{-3}$, respectively. This indicates the correct response of TDR sensors. The higher the water dose, the higher the increase in volumetric moisture. In

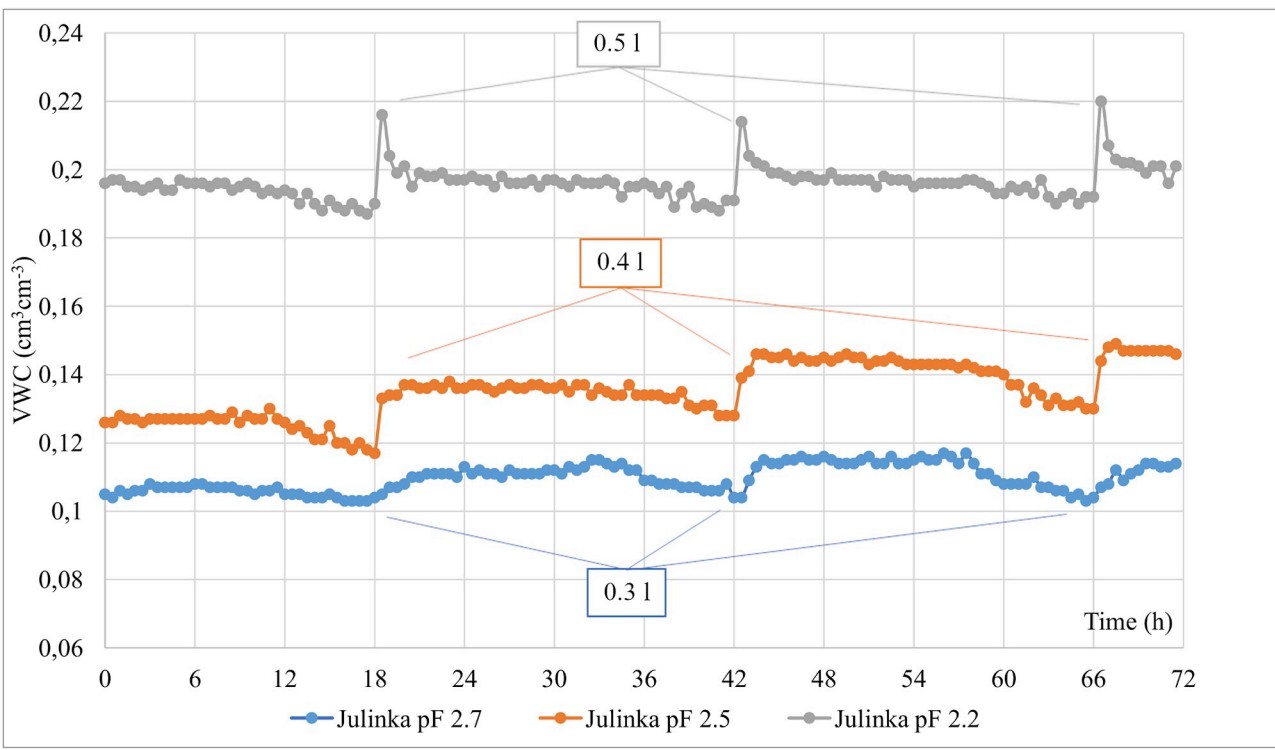

**Fig 4. Dynamics of volumetric moisture of soil in the cultivation of potato cv.** Julinka measured with TDR apparatus.

phase II, in the variant in which the water dose was 0.5 l, the volumetric moisture significantly decreased by about 0.018 $cm^3$ $cm^{-3}$. The situation was different in the other variants in which the volumetric moisture was essentially at a constant level. In phase III, the decreases were similar and amounted to about 1.5%. The above observations constitute a premise for the correctness of the response of the regulated system to the controlling factor.

The proportionality of the intensity of soil volumetric humidity changes with increases in the applied water dose and this is also noticeable using simulations based on mathematical models [38,39]. Such models are most often developed using the Richards equation and van Genuchten parameters [39–42]. Graphic presentations clearly indicate that the application of an increasing dose of water into the soil causes an increased intensity of moisture changes [39].

In addition, to assess the accuracy of the measurements carried out, the results of volumetric moisture measurements performed using TDR sensors and the SM150-Kit probe were compared using the classic statistical measures. The results of the analyses are presented in Table 2.

The averages as well as the medians showed greater convergence with the assumed values when the measurement was performed with the SM150-Kit probe. This is justified because, based on the values read, the amount of hydration was determined in accordance with the adopted procedure. The TDR probe measurement showed lower values–even in the range of several percentage points. This difference resulted from the different location of the probe sensor and from the greater amount of data collected. Irrespective of the adopted measurement method and potato variety, identical trends in coefficients of variation were obtained. Keeping soil moisture at an increased level gave it greater stability. The measurements made with TDR sensors provided values at a very low level (3.1 and 4.3%, for Denar and Julinka

**Table 2. Classic statistical measures for the measurement results obtained with the use of TDR and SM150-Kit sensors.**

| Variety | pF | Determined humidity ($cm^3cm^{-3}$) | Sensor SM150-Kit measurement | | | | TDR apparatus measurement | | | |
|---|---|---|---|---|---|---|---|---|---|---|
| | | | average ($cm^3cm^{-3}$) | median ($cm^3cm^{-3}$) | standard deviation ($cm^3m^{-3}$) | coefficient of variation (%) | average ($cm^3cm^{-3}$) | median ($cm^3cm^{-3}$) | standard deviation ($cm^3m^{-3}$) | coefficient of variation (%) |
| Denar | 2.7 | 0.198 | 0.188 | 0.175 | 0.054 | 28.8 | 0.092 | 0.069 | 0.046 | 50.2 |
| | 2.5 | 0.223 | 0.195 | 0.206 | 0.050 | 25.7 | 0.145 | 0.167 | 0.067 | 46.1 |
| | 2.2 | 0.272 | 0.246 | 0.247 | 0.038 | 15.6 | 0.192 | 0.191 | 0.006 | 3.1 |
| Julinka | 2.7 | 0.198 | 0.155 | 0.173 | 0.063 | 40.6 | 0.066 | 0.061 | 0.025 | 41.2 |
| | 2.5 | 0.223 | 0.191 | 0.198 | 0.051 | 26.9 | 0.128 | 0.136 | 0.027 | 20.2 |
| | 2.2 | 0.272 | 0.248 | 0.245 | 0.044 | 17.9 | 0.201 | 0.202 | 0.009 | 4.3 |

varieties, respectively). Maintenance of a lowered level of soil moisture was the most difficult task and was prone to strong fluctuations. Larger differences were found when using TDR sensors (50.2% and 41.2% for Denar and Julinka, respectively). The above differences may have been caused primarily by the different locations of the sensor rods, their different lengths and the different spacings between them. This, however, is a hypothesis that needs to be verified.

## 3.2. Irrigation process

The irrigation criterion was the volumetric moisture of soil in the pots. The moisture content required for a given variant was read from the pF curve prepared for sandy loam. Three types of curves proposed by Zawadzki (1999) [40] Hewelke (2015) [41] and van Genuchten (1980) [42] were analysed. Table 3 presents the values of volumetric moisture content for pF values corresponding to the 3 variants. Since the pF value for each type of curve is different depending on $\theta$ ($\theta$- volumetric moisture), mean values were used for further calculations. In the experiment, irrigations were conducted in such a way that volumetric moisture in the pots was maintained at a level of moisture corresponding to low water content ($\theta_{low} = 0.198$ $cm^3$ $cm^{-3}$) for pF 2.7, optimal moisture content ($\theta_{opt} = 0.223$ $cm^3$ $cm^{-3}$) for pF = 2.5, and a variant with high water content ($\theta_{high} = 0.272$ $cm^3$ $cm^{-3}$) for pF = 2.2. There were 30 pots of each given potato variety per irrigation variant.

$\theta_{low}$, $\theta_{opt}$, $\theta_{high}$ – low, optimal and high moisture required in three variants

$\Theta$ – calculated mean values

In the experiment, the controlled value was the volumetric moisture of the substrate with the root system for three variants. The regulating variable–decisive for the controlled value–was the amount of the water dose applied and the frequency of irrigation. In addition to the regulating variable, the controlled value was also affected by interfering factors, i.e. water uptake by plant roots (desuction) and evaporation from the open soil surface. A diagram of the procedure for decision-making regarding the irrigation parameters is presented in Fig 5.

**Table 3. Volumetric moisture content for the established pF values.**

| References | Value $\theta_{low}$ for 2.7 pF ($cm^3$ $cm^{-3}$) | Value $\theta_{opt}$ for 2.5 pF ($cm^3$ $cm^{-3}$) | Value $\theta_{high}$ for 2.2 pF ($cm^3$ $cm^{-3}$) |
|---|---|---|---|
| Saturnin Zawadzki, 1999 [40] | 0.215 | 0.240 | 0.275 |
| Hewelke, Piotr et al., 2015 [41] | 0.220 | 0.250 | 0.320 |
| Van Genuchten, M. Th., 1980 [42] | 0.160 | 0.180 | 0.220 |
| Average moisture | $\bar{\theta}_{low} = 0.198$ | $\bar{\theta}_{opt} = 0.223$ | $\bar{\theta}_{high} = 0.272$ |

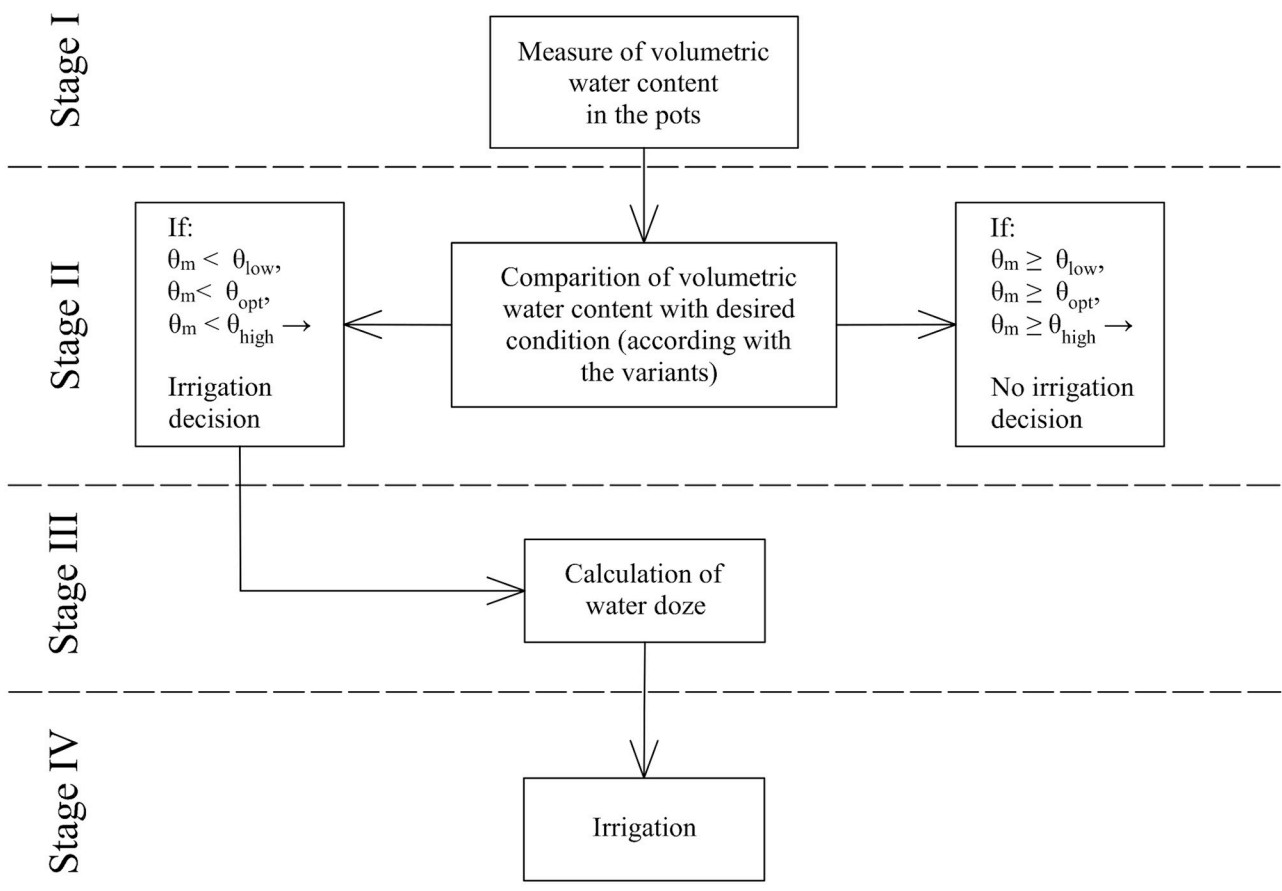

**Fig 5. Diagram of the procedure for making irrigation decisions.**

In the first stage while making irrigation decisions, the volumetric soil moisture in the pots was measured with the use of the SM150-Kit sensor.

$\theta_{low}$, $\theta_{opt}$, $\theta_{high}$ – low, optimal and high moisture content required in three variants

$\theta_p$ – measured moisture content

In stage 2, during making irrigation decisions, the measured values were compared with the required moisture for the selected variant set out in Table 3. If the measured moisture was lower than the required moisture, a decision on irrigation was made. In stage 3, the amount of water applied was calculated for the variant with the optimum water content. The following formula (1) was used [43]:

$$D_{j,opt} = (\theta_{opt} - \theta_p) \cdot V_p, \tag{3}$$

where:

$D_{j,opt}$ – unit dose of applied water (liter),

$\theta_{opt}$ – average volumetric moisture content calculated for the adopted variants (Table 1) (cm$^3$ cm$^{-3}$, %),

$\theta_p$ – average volumetric moisture content from 30 pots calculated on the basis of SM150-Kit measurements (cm$^3$ cm$^{-3}$, %),

$V_p$ – volume (liter).

Analogically, the unit irrigation dose for cases of water deficit ($D_{j,\text{low}}$) and excess water ($D_{j,\text{high}}$) was calculated.

The SM150-Kit sensor allows measurement at multiple points without destroying the root system. Therefore, this technique was used to determine the unit irrigation rate. Dose calculations and irrigations were carried out twice a week. The volume of applied water was calculated for each potato variety and each pF level (2.2, 2.5, 2.7) using formula 1 (Table 4). The calculated values of water doses were related to one-day periods. Changes to the daily volume of the applied water took place at subsequent measurements of volumetric moisture carried out with the SM150-Kit sensor.

Measurement with this type of sensor was also helpful in other research aiming to determine the impact of biochar and composting processes on the reduction of stress and increase in yields of maize cultivation. Volume moisture measurements were performed before making the decision to start irrigation in order to implement a differentiating factor in the form of different levels of water stress during plant cultivation [44,45].

The irrigation pattern of Julinka and Denar varieties showed a varietal differentiation (Fig 6). Despite the fact that the criterion for making decisions concerning the determination of the unit dose was the same for each of the varieties, the doses were differentiated and higher for the Julinka variety.

The total volume of water used for irrigation of the cultivars is presented in Fig 7. This information forms the basis for the assessment of irrigation efficiency in terms of yield and optimisation of air and water conditions in soil.

Research on water consumption in potato cultivation has also been conducted in other research [46]. Water consumption for a single pot in the examined experiment amounted to less than 17 liters. This result is comparable with irrigation levels corresponding to pF 2.7 in our own research concerning another variety of potato (Folva), cultivated on another type of soil (sand) under specific conditions (temperature: 20/14 ± 2˚C; photoperiod—15 h) [46].

## 3.3. Evaluation of biometric characteristics of the tested potato varieties

The conducted research revealed differences between biometric features of the potato varieties. The results showed that the Denar variety had significantly longer shoots, whereas Julinka had a significantly higher weight of roots, stolons, tubers and of single tubers (Table 5). It was proved that, regardless of the soil moisture level, the Denar variety, in comparison with Julinka, on average by formed 38% longer aboveground shoots, while their weight remained statistically different. The Denar variety was characterised by a lower mass of roots, stolons and tubers: 32.2%, 49% and 25.4% lower, respectively. The Denar variety had a higher share of stems in the total plant weight and at the same time a lower share of tubers than did the Julinka variety (Fig 8).

Table 4. Example of a daily irrigation dose calculated using formula 3 ($D_j$).

| Variety water regime | Beginning humidity ($cm^3 cm^{-3}$) | Humidity for pF ($cm^3 cm^{-3}$) | Water doses (liter) |
|---|---|---|---|
| Denar pF 2.2 | 0.239 | 0.272 | 0.4 |
| Denar pF 2.5 | 0.161 | 0.223 | 0.7 |
| Denar pF 2.7 | 0.123 | 0.198 | 0.9 |
| Julinka pF 2.2 | 0.223 | 0.272 | 0.6 |
| Julinka pF 2.5 | 0.127 | 0.223 | 1.2 |
| Julinka pF 2.7 | 0.096 | 0.198 | 1.2 |

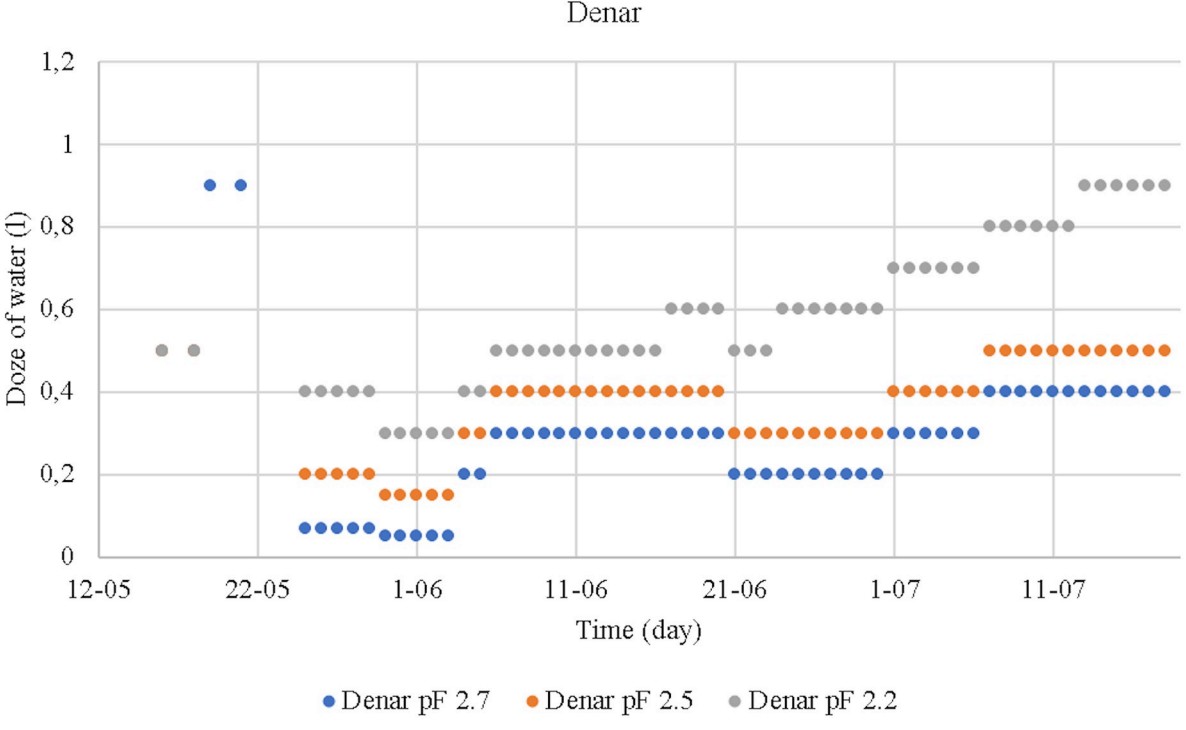

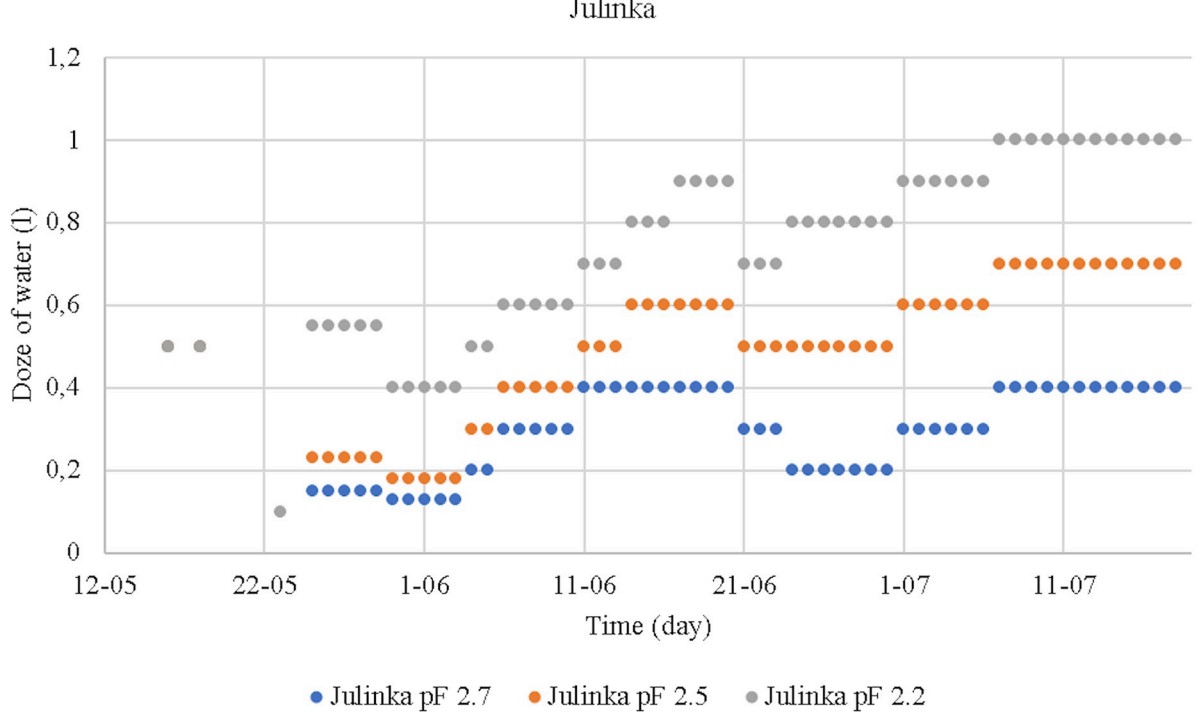

**Fig 6. Irrigation unit doses (liter).**

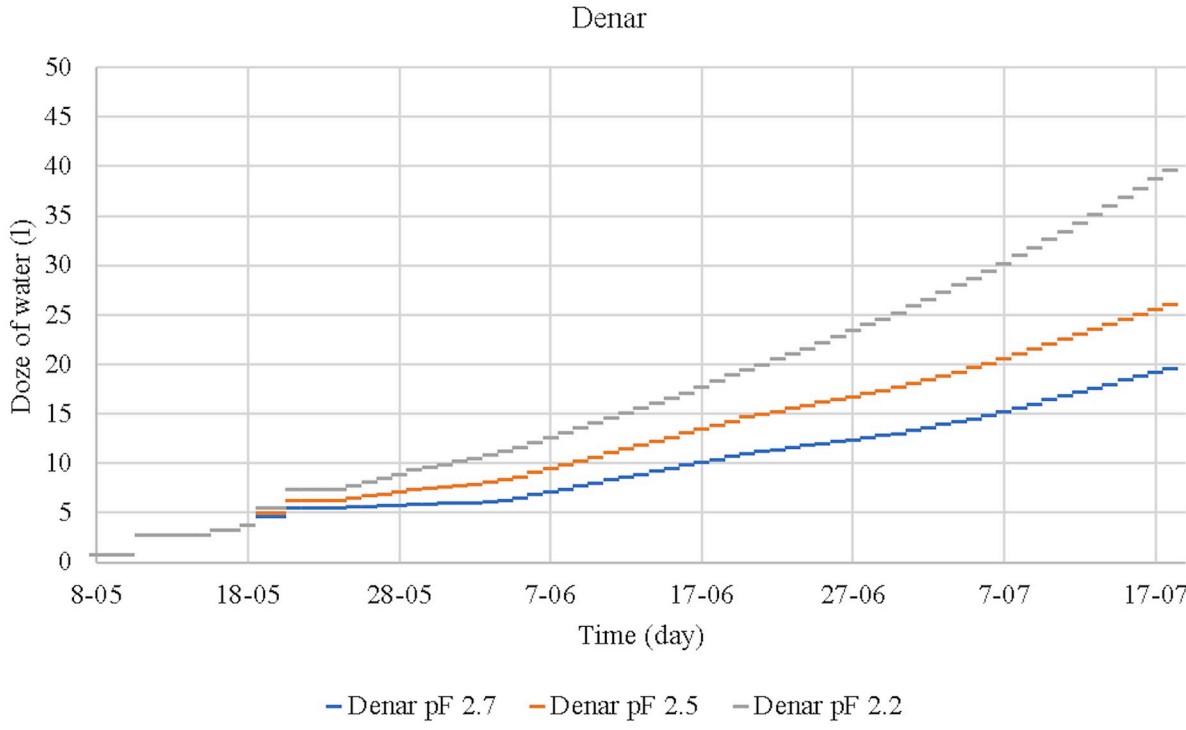

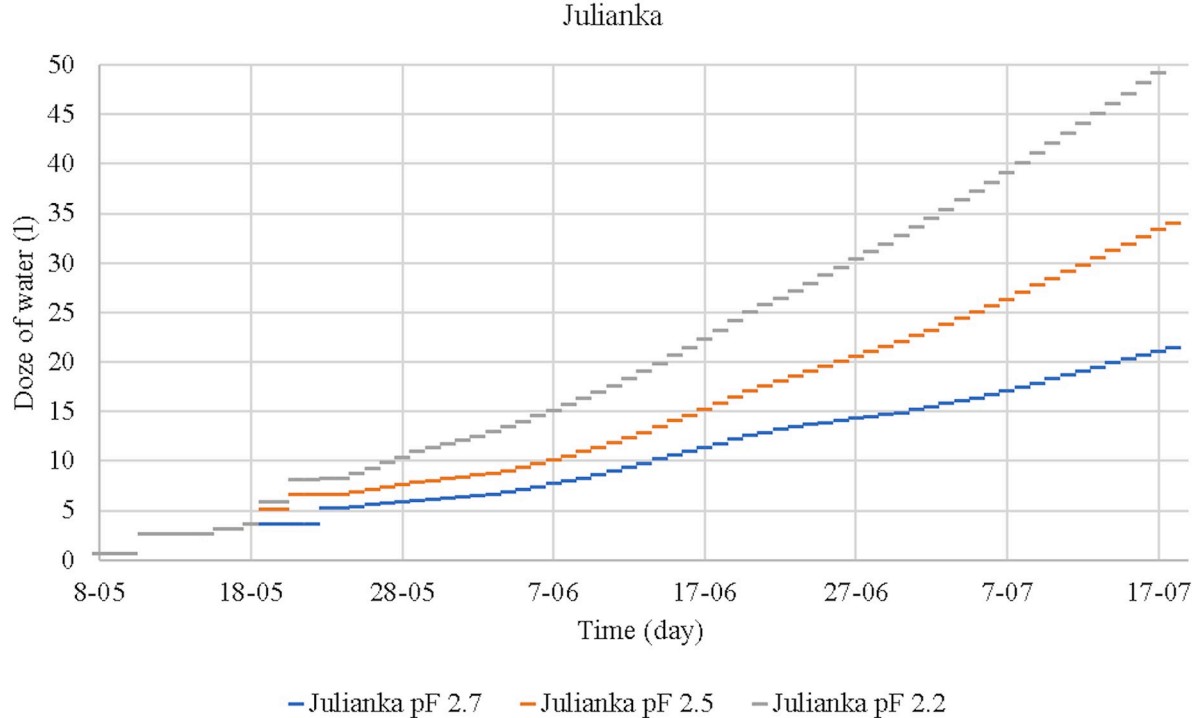

**Fig 7. Total water consumption for irrigation purposes (liter).**

**Table 5. Biometric features of potato depending on the examined factors.**

| Factor | | Number of steam (pcs) | Length of steam (cm) | Weight of steam | Weight of roots | Weight of stolon | Weight of tuber | Weight of single tuber |
|---|---|---|---|---|---|---|---|---|
| | | | | | | (g) | | |
| Variety (I) | | | | | | | | |
| Denar | | 5 | 82.17a | 148.06 | 7.97b | 0.69b | 166.67b | 18.59b |
| Julinka | | 5 | 60.50b | 161.60 | 11.77a | 1.35a | 223.51a | 28.14a |
| Water irrigation regime (II) | | | | | | | | |
| pF 2.7 | | 5 | 79.08 | 209.04a | 13.23a | 1.11 | 223.53 | 25.90 |
| pF 2.5 | | 6 | 71.67 | 155.33ab | 9.05ab | 1.17 | 213.06 | 24.53 |
| pF 2.2 | | 4 | 63.25 | 100.12b | 7.32b | 0.77 | 148.68 | 19.67 |
| Interaction (IxII) | | | | | | | | |
| Denar | pF 2.7 | 4 | 82.50a | 143.52ab | 7.42b | 0.72 | 143.48b | 18.90 |
| | pF 2.5 | 5 | 82.50a | 161.76ab | 9.24b | 0.76 | 196.72ab | 20.73 |
| | pF 2.2 | 5 | 81.50a | 138.91b | 7.24b | 0.58 | 159.82b | 16.15 |
| Julinka | pF 2.7 | 6 | 75.67ab | 274.56a | 19.04a | 1.51 | 303.59a | 32.91 |
| | pF 2.5 | 6 | 60.83ab | 148.91ab | 8.87b | 1.58 | 229.41ab | 28.33 |
| | pF 2.2 | 4 | 45.00b | 61.32b | 7.40b | 0.97 | 137.55b | 23.19 |

It was shown that, at the lower and optimal levels of soil moisture (pF 2.7, 2.5), the weight of potato stems and roots, regardless of the variety, was at the same level of significance. However, a tendency for in an increase in weight was observed at the lower moisture content. In the sites with the highest soil moisture content, stem weight decreased by 52.1% and root weight by 44.7% in comparison with plants from sites with soil moisture at the level of pF 2.7.

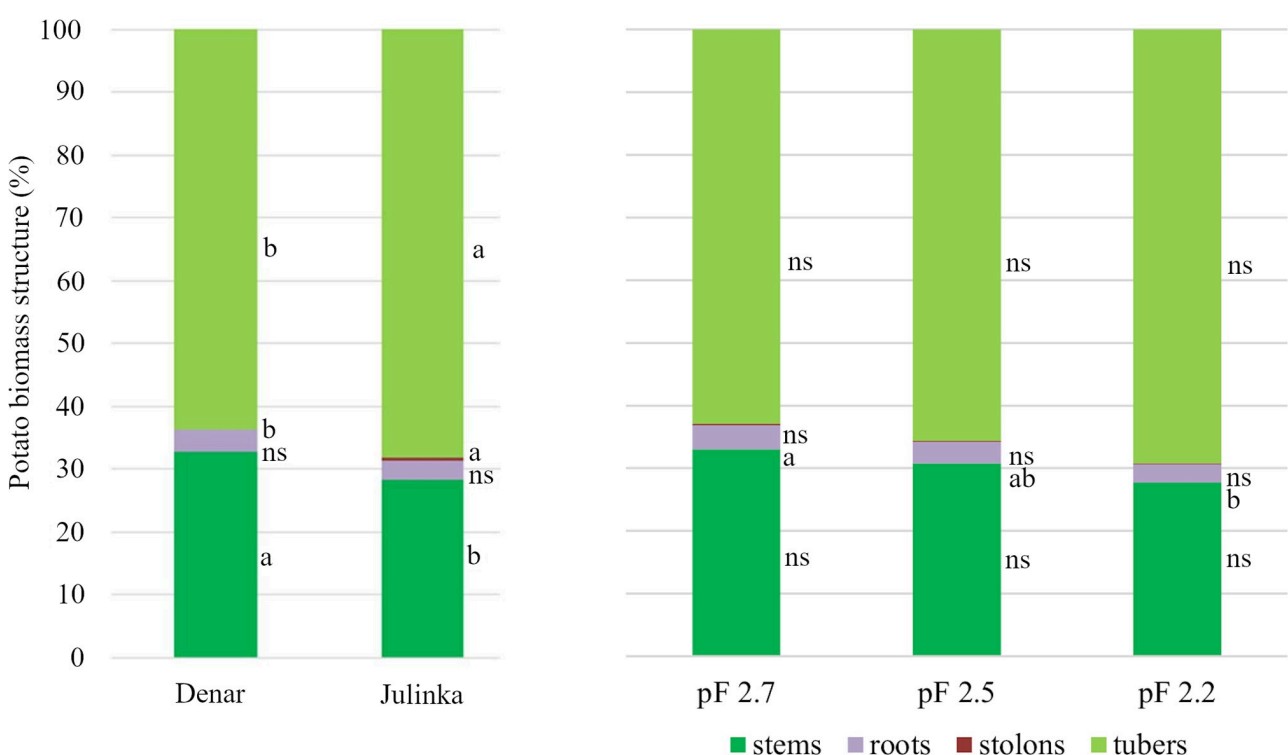

**Fig 8. Percentage of plant components depending on the variety and irrigation intensity.**

The roots contributed the lowest share to the total plant weight. No influence of soil moisture level on other biometric features of plants was demonstrated.

Statistical analysis of the interaction between the examined factors indicates a different response of the potato varieties to soil moisture. Julinka proved to be more sensitive to variations in soil moisture. A tendency for a shortening of its shoots and a decreasing of the weight of stems, roots and tubers was observed in these plants under the influence of soil moisture increases. It was statistically proved that, in sites with pF 2.2, plants formed 77.7% less stem weight, 61.1% less root weight and 56.7% less tuber weight than in sites with pF 2.7. In the case of the Denar cultivar, the influence of variable substrate moisture on biometric features was not statistically proved. However, a tendency to create a greater mass of stems and tubers was found while maintaining optimal soil moisture. The differences between the highest and the lowest values of these traits for the Denar variety were estimated at 14.1% and 27.1%, respectively.

In the studies conducted by Ojala et al. [47] and Shock and Feibert [48], potato tuber yields decreased with increasing soil moisture, similar to our study. High soil moisture contributed to a decrease in the yield and quality of tubers (tubers with reduced unit weight). Potato varieties differ in their tolerance to water stress and some can be cultivated with limited irrigation, which does not impair tuber characteristics. The use of more efficient irrigation methods in combination with the use of soil moisture monitoring systems may enable the introduction of solutions of deficit irrigation of potato. A study conducted by Erdem et al. [49] did not show any significant effect of drip irrigation on tuber yield, plant height or other yield parameters (size, length, weight and number of tubers per plant). Onder et al. [50] found that drip irrigation, surface irrigation and sub-surface irrigation methods did not significantly affect tuber yield under the soil and climatic conditions of Turkey. Other studies report increased tuber yield using drip irrigation [5,42,45–47]. Excessive irrigation of plants not only increases water and energy consumption (higher inputs), but can also lead to soil erosion, causing the leaching of fertilisers, which is an additional degradation factor [48,49].

When irrigation does not fully satisfy the needs of the plant, the yields and quality of the final product are lower and of lower quality [50–54].

The averages marked with different letters differ significantly according to the Tukey test. No markings indicate no significant differences.

## 3.4. Efficiency of irrigation treatments

First of all, when analysing the data contained in Table 6, the irrigation efficiency was assessed taking into account fresh and dry plant weight. High/ variability of irrigation effects was found. Maintaining the moisture content at a lower constant level during the vegetation

**Table 6. Irrigation efficiency.**

| Variety | pF | Fresh biomass | | Dry biomass | |
|---------|-----|---------------------------|-----------------------------|---------------------------|-----------------------------|
| | | ml water per 1 g of tuber | ml water per 1 g biomass | ml water per 1 g of tuber | ml water per 1 g biomass |
| Denar | 2.7 | 136.6 | 66.4 | 954.4 | 590.4 |
| | 2.5 | 132.4 | 70.7 | 856.7 | 563.0 |
| | 2.2 | 247.8 | 129.2 | 1617.9 | 1035.3 |
| Julinka | 2.7 | 70.8 | 35.9 | 452.3 | 284.0 |
| | 2.5 | 148.4 | 87.6 | 964.5 | 634.8 |
| | 2.2 | 364.6 | 242.0 | 2198.0 | 1646.5 |

period, regardless of the variety, contributed to the most effective use of water in the Julinka variety.

For the Denar variety, ensuring optimum moisture content was the most efficient irrigation and the water consumption per 1 g of fresh and dry matter of tuber was the lowest. The same response from the varieties was found in relation to an increase in substrate moisture. In the Julinka variety, the difference was greater and excessive moisture resulted in water consumption per 1 g of tubers of over 5 times higher than in the case of low moisture. In the Denar variety, the increase was lower, 1.8 times so. The analysis showed that in each of the analysed cases the effects depended significantly on the potato cultivar.

Then, the values of the indicator calculated in accordance with formula 2, were analyzed. The total amount of water used for irrigation purposes, and tuber mass, as well as the WUE indicator based on these two parameters are presented in Table 7. The calculated values of the *WUE* index range from 2.74 to 13.82 g l$^{-1}$ (Table 7). The standard deviation of the *WUE* harvest is $\varsigma = 4.31$ and the coefficient of variation totals $\upsilon = 1.62$. This indicates a significant variability for the individual elements of the collection of values. This is also evidenced by the quotient of the highest irrigation efficiency index to the lowest, $\varepsilon \approx 5.04$. This character of the results is caused by the fact that the experiment maintains different volumetric moistures in the soil. In the case of the Denar variety, a previously established hypothesis was confirmed, i.e. that a level of moisture corresponding to pF = 2.5 should be characterized by the highest *WUE* index. In turn, for the Julinka cultivar the results contradict this thesis. The irrigation efficiency for the underwater variant (pF = 2.7) was twice as good as that for pF = 2.5 and as much as five times better than that for pF = 2.2.

According to Liu et al. (2006) [55], the *WUE* index was determined at variable irrigation levels of potatoes grown on soil defined as sandy soil. There were three irrigation variants in this experiment: "deficit irrigation" (DI), "partial root-zone drying" (PDI) and "full irrigation" (FI). *WUE* values for these levels were as follows: 5.40, 4.98 and 5.01 g l$^{-1}$ respectively, for the given variants. In the case of the cited studies the totals $\varepsilon \approx 1.08$, which indicates a lack of discrepancy in results. These three variants of irrigation level also constituted a differentiating factor in Wang et al. [56]. The *WUE* indicator for DI was in the range of 6.28 ± 0.66 g l$^{-1}$. In turn, for PDI it was 6.08 ± 0.34 g l$^{-1}$. The lowest irrigation efficiency index was obtained for FI, which was in the range of 4.85 ± 0.48 g l$^{-1}$. The result of the research was an indicator level of $\varepsilon \approx 1.59$.

Two levels of irrigation in potato cultivation were also examined according to the experiment conducted at Massey University [57]. Irrigation variants were referred to the evapotranspiration index (ET) that was measured during the extended experiment comprising 60% ET and 100% ET levels. The set of elements in the form of *WUE* indicators for the fertilization level of 50 kg N·ha$^{-1}$ are characterized by statistical measures ($\varsigma = 2.92$, $\upsilon = 3.38$, $\varepsilon = 3.31$) and for the level of 200 kg N·ha$^{-1}$ the quoted measures are: $\varsigma = 2.81$, $\upsilon = 3.94$, $\varepsilon = 2.61$. These results are comparable measures to those obtained in the experiment carried out in our own research.

Table 7. Irrigation efficiency–*WUE* (water use efficiency) indicator.

| Variety water irrigation regime | Tuber mass (per one pot) | Volumen of used water (per one pot) | *WUE* |
|---|---|---|---|
| | (kg) | (liter) | (g l$^{-1}$) |
| Denar pF 2.7 | 0.143 | 20 | 7.15 |
| Denar pF 2.5 | 0.197 | 26 | 7.58 |
| Denar pF 2.2 | 0.159 | 40 | 3.98 |
| Julinka pF 2.7 | 0.304 | 22 | 13.82 |
| Julinka pF 2.5 | 0.229 | 34 | 6.74 |
| Julinka pF 2.2 | 0.137 | 50 | 2.74 |

**Table 8. Irrigation efficiency indicators $E_f$ (g ml$^{-1}$).**

| Variety water irrigation regime | Denar | Julinka |
|---|---|---|
| pF 2.2 | 5.09 | 2.00 |
| pF 2.5 | 5.15 | 2.76 |
| pF 2.7 | 6.85 | 2.85 |

Considering the above research, the diversity of the *WUE* index obtained in our own research is not unusual. The multitude of factors affecting potato yield means that, despite an attempt to maintain weather and soil conditions at a constant level, this is still a complex issue.

**3.4.1. Dynamics of volumetric moisture content in relation to irrigation doses applied.**   Another way to present the irrigation efficiency was to analyse the value of the coefficient determined by formula 3. The results of calculations are presented in Table 8. In both varieties, the highest, i.e. the most beneficial, irrigation efficiency index is noted for variants maintaining the moisture content at the level of pF 2.7. This results from the fact that this is the lowest moisture level (0.198 cm$^3$ cm$^{-3}$) and applications of any doses of water gave, relatively, the highest increase. It is also worth noting that the difference in efficiency indicators between the potato varieties is more than two fold.

# 4. Conclusions

The water requirements of the two potato varieties analysed here are different, increasing with the growth and development of the aboveground and underground parts. It is necessary to accumulate water for the later vegetation periods or to cultivate in areas where rainfall distribution will ensure economically viable yields. The methods used in the study to monitor soil moisture and the adopted schedule of procedures showed that they are suitable for maintaining soil moisture at the assumed level. The applied irrigation doses based on soil moisture assessment using the SM150-Kit probe contributed to differentiated soil moisture in the tuber formation zone. Moisture assessment using SM150-Kit probe was always lower than the assumed level.

Irrigation of both cultivars was characterized by significant diversity. The highest daily level of irrigation was noted for the Denar cultivar (irrigation variant pF 2.7), amounting to 0.9 l, and for the Julinka cultivar (irrigation variant pF 2.5 and pF 2.7): 1.2 liter. It was demonstrated that the water requirements of both potato varieties are different and increase with the development of the aboveground and underground parts. Moreover, it was shown that irrigation needs of the Julinka cultivar were 31.4 to 33% higher than those of the Denar variety, depending on the variant adopted. Regardless of the cultivar and water irrigation regime, the tubers had the largest share in the yield structure. The weight of tubers, as well as of a single tuber, was significantly dependent on the cultivar. The tuber mass per plant was 223.51 and 166.67 g, respectively, for Denar and Julinka cultivars, while single tuber mass was 28.14 and 18.59 g, respectively. The usefulness of the applied methods of soil moisture monitoring to prepare an irrigation schedule for the determination of irrigation doses so as to ensure adequate air-water conditions of the substrate was confirmed. The greatest value for Water Use Efficiency was obtained for the Julinka variety–the highest mass of tubers with a low amount of irrigated water per pot. The highest *WUE* index was observed for the Julinka cultivar (pF 2.7)– 13.82 g l$^{-1}$– with the highest weight of obtained tubers. The research showed that the most effective method of potato cultivation is to maintain soil moisture at a lower level. This should be taken

into account in areas where the cultivation of this species will be carried out with supplementation of water needs by irrigation.

## Author Contributions

**Conceptualization:** Amadeusz Walczak, Katarzyna Adamczewska-Sowińska, Grzegorz Janik, Józef Sowiński, Grzegorz Pęczkowski.

**Data curation:** Amadeusz Walczak, Lilianna Głąb, Xinhao Chen.

**Formal analysis:** Amadeusz Walczak, Grzegorz Janik, Izabela Kłosowicz, Lilianna Głąb, Józef Sowiński, Xinhao Chen.

**Investigation:** Anna Jama-Rodzeńska, Izabela Kłosowicz, Lilianna Głąb, Józef Sowiński, Xinhao Chen.

**Methodology:** Anna Jama-Rodzeńska, Amadeusz Walczak, Grzegorz Janik, Józef Sowiński, Grzegorz Pęczkowski.

**Project administration:** Grzegorz Pęczkowski.

**Resources:** Amadeusz Walczak.

**Software:** Izabela Kłosowicz, Lilianna Głąb, Grzegorz Pęczkowski.

**Supervision:** Katarzyna Adamczewska-Sowińska, Grzegorz Janik, Józef Sowiński, Grzegorz Pęczkowski.

**Validation:** Amadeusz Walczak, Katarzyna Adamczewska-Sowińska, Izabela Kłosowicz.

**Visualization:** Amadeusz Walczak, Katarzyna Adamczewska-Sowińska.

**Writing – original draft:** Amadeusz Walczak, Katarzyna Adamczewska-Sowińska.

**Writing – review & editing:** Anna Jama-Rodzeńska, Grzegorz Janik, Józef Sowiński, Grzegorz Pęczkowski.

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
