## [Decision Letter · Decision Letter 0]

24 Feb 2020

PONE-D-19-34935

Influence of variation in the volumetric moisture content of the substrate on irrigation efficiency in early potato varieties

PLOS ONE

Dear Dr Jama-Rodzeńska,

Thank you for submitting your manuscript to PLOS ONE. After careful consideration, we feel that it has merit but does not fully meet PLOS ONE’s publication criteria as it currently stands. Therefore, we invite you to submit a revised version of the manuscript that addresses the points raised during the review process.

We would appreciate receiving your revised manuscript by Apr 09 2020 11:59PM. To enhance the reproducibility of your results, we recommend that if applicable you deposit your laboratory protocols in protocols.io, where a protocol can be assigned its own identifier (DOI) such that it can be cited independently in the future. For instructions see: http://journals.plos.org/plosone/s/submission-guidelines#loc-laboratory-protocols

We look forward to receiving your revised manuscript.

Kind regards,

Vassilis G. Aschonitis

Academic Editor

PLOS ONE

Journal Requirements:

1. We note you have included a table to which you do not refer in the text of your manuscript. Please ensure that you refer to Table 4 in your text; if accepted, production will need this reference to link the reader to the Table.

Reviewers' comments:

Reviewer's Responses to Questions

**Comments to the Author**

1. Is the manuscript technically sound, and do the data support the conclusions?

Reviewer #1: Partly

Reviewer #2: Partly

2. Has the statistical analysis been performed appropriately and rigorously? 

Reviewer #1: No

Reviewer #2: Yes

3. Have the authors made all data underlying the findings in their manuscript fully available?

Reviewer #1: Yes

Reviewer #2: Yes

4. Is the manuscript presented in an intelligible fashion and written in standard English?

Reviewer #1: Yes

Reviewer #2: Yes

5. Review Comments to the Author

Reviewer #1: Introduction

What’ the novelty of this research?

Materials and Methods

Lines 96-97: why the two early varieties were used in the pot experiment?

Line 100: no fertilizer was applied into pots? Why?

Line 106: What’s hydraulic and physical parameters? What’s nutrient content of the soil?

Lines 123-124: move this sentence to Results and Discussion.

Lines 122-199: this section is lengthy. Several sentences should be moved to Results and Discussion.

Line 216: change kg l-1 to kg m-3.

Line 225: Are there irrigation water leaching out pots?

Line 227: change the unit of EF to %.

Results and Discussion

This section was too lengthy, and it is difficult to follow. There were too much words used to describe the experimental results, little words used to discuss the results compared with previous studies.

Conclusions

Lines 413-417: delete these sentences.

Reviewer #2: The manuscript in its present format needs a revision of the accuracy data present to interpret the results to support the validity of the experimental findings. Specific comments are as follows:

Line 95-97: The authors use two different potato varieties. I think it is not correct, to compare and show the results of irrigation doses and water consumption of totally different potato varieties. Because of the different physiological processes in these varieties, methods of cultivation, and final purpose.

Line 136-149: Why authors give more attention to describe size, depth, etc., for SM150-Kit, in contrast, the TDR sensor?

Line 171-172: Will be used only SM150-Kit for the first stage of soil moisture measurements? What about another sensor? Then in Line 192-194: The authors will comparative both methods, please explain this?

Line 260: Should to add an explanation and a reference in the manuscript in Fig. 3?

Line 283-285: I have reservations about this. Before starting the experiment, the authors should make the calibration of both sensors and find the correlation between the data of sensors. It was not good to explain results because of “low data and different locations of sensors”.

Line 338-341: How do you think, maybe for Julinka it was not a good condition for growing because of more water and you had a result which you describe on Line 339-341?

Propose to amplify the conclusions.

6. PLOS authors have the option to publish the peer review history of their article (what does this mean?). If published, this will include your full peer review and any attached files.

Reviewer #1: No

Reviewer #2: No

---

## [Author Response · Author response to Decision Letter 0]

30 Mar 2020

Dear Editors PlosOne,

We would like to thank for noticing our study, the difficulty of reading manuscript and comments of the manuscript entitled ”Influence of variation in the volumetric moisture content of the substrate on irrigation efficiency in early potato varieties” authorship Anna Jama-Rodzeńska, Amadeusz Walczak, Katarzyna Adamczewska-Sowińska, Grzegorz Janik, Izabela Kłosowicz, Lilianna Głąb, Józef Sowiński, Xinhao Chen, Grzegorz Pęczkowski. We edit the manuscript according to your suggestions and requirements of PlosOne. All suggestions and shortcomings (inter alia, no citation of table number 4) has been completed. 

Below I would like to refer to the comments of reviewers: 

Answers on Reviewer #1: 

Introduction. What’ the novelty of this research?

Potato is one of crops deeply dependent on global climate change. Production is very risky in limited water regions influenced on quantity and yield quality. The irrigation technique and water utilization are very important factor of potato production. Central Europe Region depending on drought period more often and production of some crops (f.ex. potato) are less risky when irrigation is included in potato cultivation. In many countries sprinkler system is most popular. Because of decreasing water resources more effective irrigation system must be used especially in those crops which reacts strongly on the canopy air humidity conditions causes diseases infection. Potato belongs to those plants. 

The novelty of our research is related not only to the method of irrigation but strict water doses dependency on soil moisture. The irrigation doses were applied and adjusted to the current soil moisture during the whole growing season estimated in very short intervals (two measurement per week). Water doses differed significantly during the growing season between varieties which suggest the need to determine the water dose for different varieties and not just for crop. Our research suggests using two sensors methods for soil moisture measurement aimed to facilitate the farmers to take such a measurement during potato growing and determine the water dose based on the current soil moisture. 

Materials and Methods. Why the two early varieties were used in the pot experiment?

Two early varieties Denar and Julinka were used in the experiment intentionally. The aim of this selection was based on similar duration of the growing season, but not similar terms in which the various development phases were taken place. Very early instead early potato varieties have the highest initial ratio of biomass accumulation especially tuber and recommended for early market harvest. Use of varieties with that developmental differences allowed to determine the influence of different irrigation variants on biomass accumulation intensity. 

Line 100: no fertilizer was applied into pots? Why?

Soil for that experiment was taken from fertilized field in spring, fertile enough for that experiment (additional data presented). Additionally, fertilization was not important in this study and was therefore omitted (especially nitrogen fertilization) because plays pivotal role in the quality and quantity of potato yield which would influenced on the effects of irrigation. 

Line 106: What’s hydraulic and physical parameters? What’s nutrient content of the soil?

Information about the parameters determining the water flow in the porous medium was added to the manuscript. Soil composition was also completed (Table 1 in text). 

Table 1. Soil chemical composition used for pot experiment

Soil parameters Value (unit)

pH 7.16

salinity 155 μS/cm

Mg 20 mg/dm3

P 51 mg/dm3

Ca 420 mg/dm3

K 52 mg/dm3

Nitrates 1,31 mg/dm3

granulometric composition coarse-textured sand

Lines 123-124: move this sentence to Results and Discussion.

Sentences connected to water movement in porous media from section Materials and Methods were moved to Results and Discussion to subsection 3.1 Testing the adequacy of the soil moisture content response to irrigation (line 194-202). 

Lines 122-199: this section is lengthy. Several sentences should be moved to Results and Discussion.

Description of soil moisture measurement methods with irrigation criteria, determination of the volume moisture for the determined value of pF, the procedure for making irrigation decisions were moved from Materials and Methods to Results and Discussion subsection 3.2 Irrigation process (line 257- 306). 

Line 216: change kg l-1 to kg m-3.

The units used in the article were changed according to the suggestion of the reviewers. At the same time, we changed the values in Table 6 (old version) to Table 7 (new version of manuscript) concerning the average weight of tubers per pot. After that changes, the values of the WUE (Water Use Efficiency) index have also changed. 

The WUE unit and other units (Ef, Dj) have been unified in accordance with Publications 55, 56, 57 treating irrigation efficiency in potato cultivation under pot experiments.

Line 225: Are there irrigation water leaching out pots?

There was no water leaching out pots because the bottom of the pots was impermeable. This information was added in Experiment set-up subsection in line 98. 

Line 227: change the unit of EF to %.

The units used in the article was changed according to the suggestion of the reviewers. 

Results and Discussion. This section was too lengthy, and it is difficult to follow. There were too much words used to describe the experimental results, little words used to discuss the results compared with previous studies.

The Results and Discussion section was changed according to reviewer suggestion. The discussed of experience results with those of other authors has been omitted. This chapter has been completed with missing information and complemented by a discussion of the results. 

Conclusions

Lines 413-417: delete these sentences.

Lines 413-417 were deleted as a general thoughts and conclusions not related to the studies carried out. 

Answers on Reviewer #2: 

The manuscript in its present format needs a revision of the accuracy data present to interpret the results to support the validity of the experimental findings. Specific comments are as follows:

Line 95-97: The authors use two different potato varieties. I think it is not correct, to compare and show the results of irrigation doses and water consumption of totally different potato varieties. Because of the different physiological processes in these varieties, methods of cultivation, and final purpose.

Answer was given for Reviewer 1 (answer 2). Two varieties testing in the pot experiment was intentional.

Line 136-149: Why authors give more attention to describe size, depth, etc., for SM150-Kit, in contrast, the TDR sensor?

Information about TDR sensors was completed in the text of the manuscript.

Line 145-148. FP/mts sensors characterize with smaller body diameter, which totals 2 cm longer bars (10 cm). In the case of this sensor for dry soil, the sensitivity zone is an elliptical cylinder with a height of 10.2 cm and radii of 0.3 cm and 0.4 cm. However, when the soil is saturated, the cylinder is 11.2 cm high and the base of the wheel has a radius of 1.7

Line 171-172: Will be used only SM150-Kit for the first stage of soil moisture measurements? What about another sensor? Then in Line 192-194: The authors will comparative both methods, please explain this?

Both sensors (SM-150K and TDR) measured volumetric moisture of soil during the whole experiment. In lines 277-285 the authors imprecisely defined and introduced the term "stage". The term „stage” means following steps during making irrigation decisions. 

Line 260: Should to add an explanation and a reference in the manuscript in Fig. 3?

Figure 3 was replaced by Figure 2. Figure 2 was done on the base on results obtained from TDR sensor. Additionally, a detailed description of the graph can be found in the manuscript (line 203-223). 

Line 283-285: I have reservations about this. Before starting the experiment, the authors should make the calibration of both sensors and find the correlation between the data of sensors. It was not good to explain results because of “low data and different locations of sensors”.

Measuring techniques using TDR and SM-150Kit sensors have been recognized and used worldwide aiming at measuring soil volumetric moisture what is confirmed by hundreds of scientific works. The implementation of these measurement techniques was preceded by research aimed at comparing the results with those of the drying and weighing methods. Therefore, both methods were not calibrated. As stated by the producers, additional calibration is required only in a case of obtaining an accuracy below 0.5 %. In this work, this is not required. However, for the reliability of the analysis, classical statistical measurements have been carried out for the results obtained for TDR and SM150-Kit sensors. 

Line 338-341: How do you think, maybe for Julinka it was not a good condition for growing because of more water and you had a result which you describe on Line 339-341?

Thank you for that comments. We selected two potato varieties not having detailed information on resistance to lack of water and reaction on irrigation. That result from that research suggest to continue research for optimizing water consumption for specific varieties and irrigation technique.

Propose to amplify the conclusions.

This comment were taken into account while improving the manuscript.

---

## [Decision Letter · Decision Letter 1]

2 Apr 2020

Influence of variation in the volumetric moisture content of the substrate on irrigation efficiency in early potato varieties

PONE-D-19-34935R1

Dear Dr. Jama-Rodzeńska,

We are pleased to inform you that your manuscript has been judged scientifically suitable for publication and will be formally accepted for publication once it complies with all outstanding technical requirements.

With kind regards,

Vassilis G. Aschonitis

Academic Editor

PLOS ONE

Additional Editor Comments (optional):

Reviewers' comments:

Reviewer's Responses to Questions

**Comments to the Author**

1. If the authors have adequately addressed your comments raised in a previous round of review and you feel that this manuscript is now acceptable for publication, you may indicate that here to bypass the “Comments to the Author” section, enter your conflict of interest statement in the “Confidential to Editor” section, and submit your "Accept" recommendation.

Reviewer #1: All comments have been addressed

2. Is the manuscript technically sound, and do the data support the conclusions?

Reviewer #1: Yes

3. Has the statistical analysis been performed appropriately and rigorously? 

Reviewer #1: Yes

4. Have the authors made all data underlying the findings in their manuscript fully available?

Reviewer #1: Yes

5. Is the manuscript presented in an intelligible fashion and written in standard English?

Reviewer #1: Yes

6. Review Comments to the Author

Reviewer #1: Authors have revised the manuscript according to reviewer's comments. All comments were addressed. My suggestion is that the manuscript could be considered for acception and publicaiton in PlOS ONE.

7. PLOS authors have the option to publish the peer review history of their article (what does this mean?). If published, this will include your full peer review and any attached files.

Reviewer #1: No

---

## [Editor Report · Acceptance letter]

6 Apr 2020

PONE-D-19-34935R1 

Influence of variation in the volumetric moisture content of the substrate on irrigation efficiency in early potato varieties 

Dear Dr. Jama-Rodzeńska:

I am pleased to inform you that your manuscript has been deemed suitable for publication in PLOS ONE. Congratulations! Your manuscript is now with our production department. 

With kind regards,

on behalf of

Dr. Vassilis G. Aschonitis 

Academic Editor

PLOS ONE